# Dynamic Banzhaf: Game-Theoretic Attribution with Dynamic Feature-Wise Probabilities

## Abstract

Game-theoretic attribution methods approximate target model as a cooperative game and evaluate feature importance as payoff allocation to the input features. Most methods use well-known game-theoretic solutions such as the Shapley value because they satisfy key desirable axioms. However, the strict assumptions of game theory reduce the flexibility of explanations: in particular, most methods use fixed coalition sampling distributions, preventing the dynamic alignment of explanations to user criteria. To address this gap, we introduce Dynamic Banzhaf, a game-theoretic attribution method that optimizes the masking probability of each feature to a user-defined objective function. We provide theoretical proof on the convergence of Dynamic Banzhaf, discuss optimal probability selection, and empirically demonstrate the effect of probability adjustment on the quality of the explanations in machine learning models. Our results indicate that masking probabilities can be calibrated to improve the alignment of explanations to user criteria, highlighting the effect of dynamic probability selection in game-theoretic attribution.

## 1 Introduction

Artificial Intelligence (AI) is becoming a ubiquitous tool in many fields owing to its capacity to reflect complicated patterns in large datasets. However, this capacity is often accompanied by high model complexity, turning a model into a black box whose prediction process is difficult to interpret. In high-stakes domains like health care or finance (Caruana et al., 2015; Grath et al., 2018), interpretability is just as important as the accuracy of predictions, and model complexity hinders the practical adoption of AI in these domains. Explainable AI addresses this challenge by attaching explanations to the models (Samek, 2017; Gunning et al., 2019; Longo et al., 2024).

Among various explanation techniques, feature attribution measures the contribution of input features to a model's prediction (Ribeiro et al., 2016b; Lundberg & Lee, 2017; Sundararajan et al., 2017). In particular, local model-agnostic approaches are popular in explainable AI as they compute the input importance regardless of target model's architecture (Ribeiro et al., 2016a). Game-theoretic explainable AI is a branch of local model-agnostic attribution that approaches the explanation process as a cooperative game, where each feature $i$ is a 'player' and the 'game' $v$ is a set function that maps a subset of features (coalition $S$) to the model's output. The feature importance is equivalent to the 'payment' allocated to player $i$, which is a weighted average of $\Delta_i v(S)$, the change in $v$ caused by adding $i$ to $S$. Thus, in game-theoretic explainable AI, the weighting scheme determines the feature importance.

The Shapley value (Shapley, 1953), which performs an average over all possible permutations of features by using combinatorial weights, is frequently used in explainable AI because it satisfies several desirable axioms. It can be approximated by KernelSHAP (Lundberg & Lee, 2017), which performs a weighted linear regression on randomly sampled coalitions. KernelSHAP has been explored thoroughly in the past literature (Lundberg et al., 2018; Sundararajan & Najmi, 2020; Chen et al., 2021; Mosca et al., 2022). However, Shapley value has several issues due to its weights. Specifically, it applies much higher weights on either very small or very large subsets, ignoring the effects of mid-sized subsets. This weighting scheme leads to instabilities in various explanation tasks that has to be addressed with specific techniques like specialized sampling (Covert & Lee, 2020). It

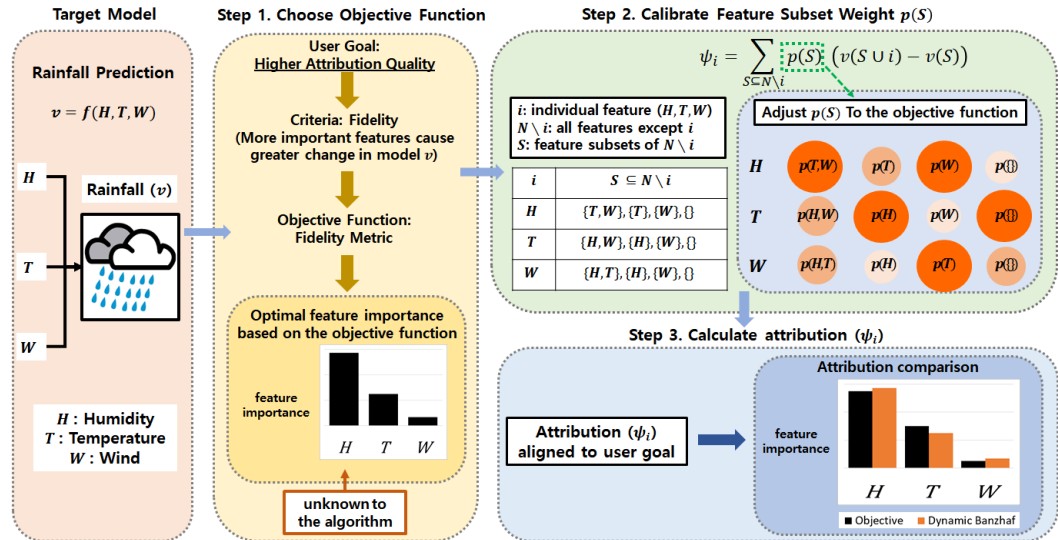

Figure 1: **Illustration of Dynamic Banzhaf value**. Given a model, a user chooses a goal for the attribution, such as improving its quality. The user defines relevant criteria and an objective function that quantifies the criteria, which leads to an optimal feature importance. These values are not known *a priori* to the algorithm (Step 1). Dynamic Banzhaf adjusts the weights of the weighted average to the objective function dynamically. Larger and darker circles represent heavier weights (Step 2). The attribution values calculated with the calibrated weights become better aligned with the optimal feature importance (Step 3).

also makes the approximation of the Shapley values numerically unstable (Karczmarz et al., 2022; Wang & Jia, 2023; Liu et al., 2024).

Another solution of cooperative games is the probabilistic values (Dubey & Weber, 1977), which take the expectation of $\Delta_i v(S)$ by using $p(S)$, the probability of coalition formation, as the weights. The Banzhaf value (Banzhaf III, 1964) is a specific probabilistic value with uniform $p(S)$, and has recently gained attention as an alternative of the Shapley value because it is still axiomatic, more intuitive, and more stable (Liu et al., 2024). It has been applied in tasks like data valuation (Wang & Jia, 2023) and feature attribution (Liu et al., 2024).

A key limitation of the Banzhaf value is that it assumes a fixed $p(S)$ that is only dependent on the number of coalitions. This strict assumption forces all coalitions to be accounted for equally in all scenarios. However, different users often prioritize different qualities in their explanations, which result in different true optimal attribution. The inflexibility of the Banzhaf value implies that it cannot adapt to these requirements. An ideal attribution method should be able to calibrate its $p(S)$ to the user's criteria. For example, consider the simple 3-input model in Figure 1, which predicts rainfall based on humidity, temperature, and wind. A user has a goal for the attribution, especially regarding improving certain desiderata. The user chooses the relevant criteria and an objective function to quantify the criteria (fidelity in the example, which is the alignment between the importance assigned to a feature and its impact on model prediction (Hedström et al., 2023; Nauta et al., 2023)). The chosen objective function implies certain optimal attribution values that are not known to the algorithm *a priori* (step 1). The feature subset weights $p(S)$ (where larger and darker circle size indicates heavier weights in the diagram) are adjusted dynamically to optimize the objective function (step 2). By using these calibrated weights, the final attribution values would be better aligned with the optimal attribution for the chosen criteria (step 3).

Based on this notion, we introduce Dynamic Banzhaf, which adjusts $p(S)$ dynamically by optimizing each feature's coalition-joining probability (or *masking* probability) to a user-defined objective function (step 2 in Figure 1). We show that the attribution values can be computed through a centered linear regression, prove the convergence rate of Dynamic Banzhaf value, discuss the masking

probability calibration process, and empirically demonstrate the benefits of masking probability calibration. Our contributions are as follows:

- We introduce Dynamic Banzhaf, a novel algorithm that efficiently computes axiom-satisfying attribution using a different masking probability for each feature that is optimized to meet user-defined attribution criteria.
- We show Dynamic Banzhaf as a linear regression with intercept centered by each feature's probability, and its theoretical convergence rate.
- We discuss the process for dynamic calculation of optimal masking probabilities, and empirically demonstrate how using Dynamic Banzhaf improves the quality of the generated explanations, highlighting the importance of optimized masking probabilities in game-theoretic attribution.

## 2 RELATED WORK

### 2.1 SHAPLEY VALUE-BASED EXPLANATION

Game theory-based explainable AI literature focuses on developing methods that satisfy axiomatic properties. They are typically based on the Shapley value (Shapley, 1953), which satisfies four axioms (linearity, dummy, symmetry, and efficiency). The Shapley value is defined as:

$$\phi_i = \frac{1}{n} \sum_{S \subseteq N \setminus i} \binom{n-1}{|S|}^{-1} [v(S \cup i) - v(S)] \tag{1}$$

Where $N$ the player set with size $n$, $S$ is a subset of players, $v(S)$ is a value function. In explainable AI, the features are equivalent to the players, and $v$ maps $S$ to model outputs.

While the Shapley value is too costly to calculate exactly, Lundberg & Lee (2017) shows that it can be estimated using a weighted linear regression, a method known as KernelSHAP. This method has been adapted in many different directions in explainable AI (Mosca et al., 2022), such as architecture specialization (Lundberg et al., 2020; Ghorbani & Zou, 2020; Wang et al., 2021) or estimation method improvements (Messalas et al., 2019; Covert & Lee, 2020). One issue with the Shapley value is that its weights are the highest for small or large coalitions, minimizing the effects of intermediate-sized coalitions. This weighting scheme also makes Shapley approximations numerically unstable algorithmically (Wang & Jia, 2023; Liu et al., 2024). Recent works relax some of the axioms to address these shortcomings. For example, Kwon & Zou (2022) propose Beta Shapley, which adjust the Shapley averaging scheme based on a Beta distribution.

### 2.2 BANZHAF VALUE-BASED EXPLANATION

Another solution of cooperative game theory is the probabilistic values (Dubey & Weber, 1977), which takes the expectation of the marginal contributions with $p(S)$, the probability of coalition formation, as the weights:

$$\phi_i = \sum_{S \subseteq N \setminus i} p(S)[v(S \cup i) - v(S)] \tag{2}$$

The Banzhaf value (Banzhaf III, 1964) is a specific probabilistic value with uniform $p(S)$:

$$\phi_i = \frac{1}{2^{n-1}} \sum_{S \subseteq N \setminus i} [v(S \cup i) - v(S)] \tag{3}$$

Intuitively, the Banzhaf value is the expected marginal contribution assuming all players may join a coalition with independent probability of $w = 0.5$. The Banzhaf value has recently gained attention as an alternative of the Shapley value because it is still axiomatic, more intuitive, and more stable

(Liu et al., 2024). Furthermore, the two values are similar qualitatively, especially in terms of the contribution ranks (Freixas et al., 2012; Karczmarz et al., 2022). KernelBanzhaf Liu et al. (2024) approximates the Banzhaf values using linear regression with mask values set to {-0.5, 0.5}; Karczmarz et al. (2022) uses the Banzhaf value for data valuation; Patel et al. (2021) uses the Shapley and the Banzhaf value to select the optimal vocabulary subset for NLP tasks; and Chhablani et al. (2024) utilizes the Banzhaf value to create counterfactuals in graph neural networks. Li & Yu (2024) generalizes the Banzhaf value to *weighted* Banzhaf value, which sets $p(S) = w^{|S|}(1-w)^{(n-|S|)}$ for data valuation. They show that optimal $w$ is dependent on the dataset and model. However, there has not been any research on computing the Banzhaf values when all features have different weights in a kernelized manner, or analytic methods for determining optimal $w$.

## 3 METHOD

### 3.1 DEFINITION

Given a set of features $N$ of size $|N| = d$, coalition $S \subseteq N$, and value function $v(S)$, let $w_i$ be the probability that feature $i$ joins $S$. Then, the Dynamic Banzhaf value $\psi_i$ of player $i$ is defined as:

$$\psi_i = \sum_{S \in N \setminus i} \left( \prod_{j \in S} w_j \prod_{j \notin S} (1 - w_j) \right) (v(S \cup i) - v(S)) \tag{4}$$

Intuitively, $\psi_i$ is the expected change in $v(S)$ given that the probability of coalition formation $p(S)$ follows a multivariate Bernouilli distribution with parameter $\mathbf{w} = \{w_1, w_2, ..., w_d\}$. The Banzhaf value is a special case where $w_i = 0.5 \quad \forall i$, while the weighted Banzhaf value is another special case where $w_i = \alpha \quad \forall i, \quad 0 \le \alpha \le 1$. In explainable AI, $w_i$ is called a *masking* probability since a feature is 'removed' from a coalition by masking it with other values.

### 3.2 APPROXIMATION OF DYNAMIC BANZHAF VALUE WITH CENTERED LINEAR REGRESSION

Once the set of probabilities $\mathbf{w}$ is fixed, we can approximate Dynamic Banzhaf value with a centered linear regression:

**Theorem 1** (Dynamic Banzhaf as centered linear regression with intercept). *Given $\mathbf{z} = \{0,1\}^d \sim Ber(\mathbf{w})$, Dynamic Banzhaf values $\boldsymbol{\psi}$ is the solution of the centered linear regression:*

$$\beta_0^*, \boldsymbol{\psi} = arg \min_{\beta_0, \boldsymbol{\beta}} E_Z[(v(\mathbf{z}) - \beta_0 - \boldsymbol{\beta}^T(\mathbf{z} - \mathbf{w}))^2] \tag{5}$$

The full proof is presented in Appendix C.

It should be noted that the theorem also holds without an intercept (e.g, (Marichal & Mathonet, 2011)), which is the traditional setup for game-theoretic explainable AI. However, we specify the inclusion of intercept term because formulations without an intercept is sensitive to vertical shifts like subtraction of a baseline value. Another benefit of including the intercept is that in terms of implementation, we do not need to center $\mathbf{z}$ to approximate the Dynamic Banzhaf values since centering does not affect the feature coefficients of a linear model with intercept.

We can derive the convergence guarantee of the linear regression approximation of Dynamic Banzhaf value as follows.

**Theorem 2** (Convergence of Dynamic Banzhaf). *Let $Z_n = \{\mathbf{z}_i - \mathbf{w}\}_{i=1,...,n}$ and $V_n = \{v(\mathbf{z})\}_{i=1,...,n}$ be $n$ samples from $\mathbf{z}_i = \{0,1\}^d \sim Ber(\mathbf{w})$ and the corresponding evaluations of $v$. Let $\boldsymbol{\beta}_n^*$ be the coefficient of centered linear regression on $Z_n$ and $V_n$. Then, given constants $\delta$, $\epsilon$, and $M > 0$, $\boldsymbol{\beta}_n^*$ converges to $\psi$ with probability $1 - \delta$ (i.e., $P(|\boldsymbol{\beta}_n^* - \psi|_2 \le \epsilon) \le 1 - \delta$) for $n = \Omega(\epsilon^{-2} M^2 \sigma^2 d^3 \gamma^4 log(4/\delta))$, where $\sigma^2 = max(w_i(1-w_i))$, $\gamma^2 = \sum_{i=1}^d 1/(w_i(1-w_i))$, and $|v(\mathbf{z})| < M$.*

The full proof is presented in Appendix C. Intuitively, the theorem states that the convergence error ($\epsilon$) decreases with larger number of samples ($n$); lower maximum value function magnitude ($M$);

lower maximum variance of the feature masks ($\sigma^2$); and lower sum of feature mask precisions (($\gamma^2$). In particular, with all else held constant, the solution converges the fastest when $\sigma^2\gamma^4$ is minimized, which occurs when $w_i = 0.5 \quad \forall i$, i.e., the regular Banzhaf value.

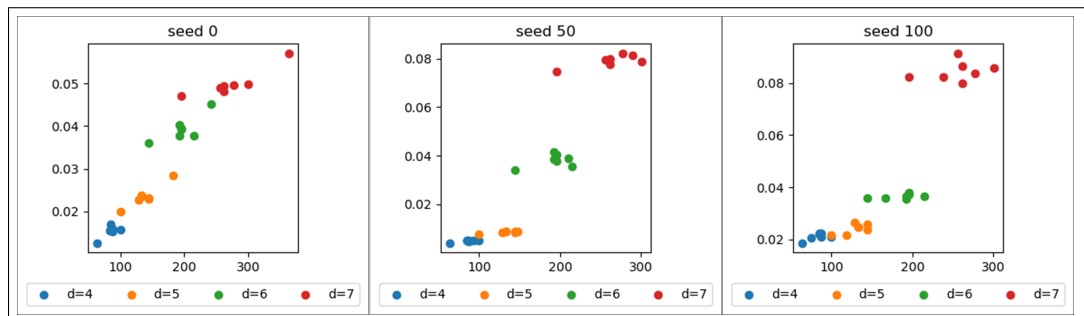

Figure 2: Convergence experiment between $L_2$ error and $\sigma^2\gamma^4$ for random masking probabilities and models generated from the specified seeds across different input dimensions. The relation is approximately linear across all seeds and input dimensions as expected from Theorem 2.

We demonstrate the linearity between $\epsilon^2$ and $\sigma^2\gamma^4$ Theorem 2 through a toy experiment (Figure 2). Each subplot presents $L_2$ error estimates versus $\sigma^2\gamma^4$ for a quadratic model with 4 to 7 input features, with model parameters, inputs and $\mathbf{w}$ randomly generated using the specified seed. All values are calculated with 1000 perturbations. The relationship is approximately linear as expected, indicating that the convergence holds.

### 3.3 CALCULATING OPTIMAL MASKING PROBABILITIES

Theorems 1 and 2 assume that masking probabilities $\{\mathbf{w}\}_{i=1,...,n}$ are known. In real world, the optimal $\mathbf{w}$ can vary widely depending on the qualities that a user expects from an attribution. These qualities are usually some desiderata of explanations (Nauta et al., 2023), such as faithfulness – the degree to which an explanation matches the model output behavior (Hedström et al., 2023) – and sensitivity – the stability of the attribution for identical inputs (Nauta et al., 2023). Once the user specifies an objective function $L(\mathbf{w})$ that quantifies the chosen quality, Dynamic Banzhaf can calibrate $\mathbf{w}$ to optimize $L(\mathbf{w})$. For continuous $L(\mathbf{w})$, we can perform gradient descent to compute $\mathbf{w}$. For some $L(\mathbf{w})$, the optimal $\mathbf{w}$ may be found theoretically: for example, for $\epsilon$ from Theorem 2 the optimal probabilities are $w_i = 0.5 \quad \forall i$.

In the experiments, we demonstrate this idea by setting $L(\mathbf{w})$ to a continuous approximation of the Area over Perturbation Curve (AOPC) (Tomsett et al., 2020), a measure of explanation fidelity that calculates the area over the curve (AOC) of the change in model output from the target input as we replace each feature in the input with the chosen baseline. The approximation is as follows:

$$AOPC_{\boldsymbol{\beta}_n^*} = \frac{1}{d+1}\sum_{k=1}^{d}(d-k+1)\boldsymbol{\beta}_{n,\pi(k)}^* \tag{6}$$

where $\boldsymbol{\beta}_{n,\pi(k)}^*$ is the $k$-th highest value in $\boldsymbol{\beta}_n^*$, the Dynamic Banzhaf value estimate from the centered linear regression. The idea is to approximate the change from replacing each feature with the corresponding estimated Dynamic Banzhaf value. The metric and its continuous approximation are discussed in Appendix A.

Since $\boldsymbol{\beta}_n^*$ is the coefficient of a weighted linear regression and therefore has an analytic solution with respect to $\mathbf{w}$, we can compute the gradient of $AOPC_{\boldsymbol{\beta}_n^*}$ using the chain rule for gradient descent optimization of $\mathbf{w}$.

## 4 EXPERIMENT

In this section, we demonstrate the effect of calibrating masking probabilities through two experiments. Firstly, we apply Dynamic Banzhaf value using approximated AOPC objective function from Section 3.3 and compare it against popular baseline methods to quantitatively evaluate the impact of dynamic masking probability adjustments on common metrics. In the second experiment, we investigate the intuitive meaning behind masking probabilities by assigning fixed probabilities across an image by location (center or periphery) and measuring the effect on explanation fidelity.

### 4.1 SETUP

**Algorithms.** We use the following algorithms for the experiments:

- KernelBanzhaf ($KBanzhaf$) (Liu et al., 2024): equivalent to setting $w_i = 0.5 \quad \forall i$.
- Weighted Banzhaf with probability $\alpha$ ($WBanzhaf(\alpha)$): equivalent to setting $w_i = \alpha \quad \forall i$. We use $\alpha$ of 0.25 and 0.75 to test the effect of uniform $\alpha$ at different levels.
- KernelSHAP (Lundberg & Lee, 2017) ($KSHAP$): linear regression approximation of Shapley value.
- Beta Shapley (Kwon & Zou, 2022) ($BetaShap(\alpha, \beta)$): this method uses the Beta distribution to compute the coalition weights of marginal contributions. We adapt the original data valuation method to feature attribution task. We use $(\alpha, \beta)$ of (16,1), (4,1), (1,4), and (1,16) following the original paper.
- Dynamic Banzhaf ($DBanzhaf$): We optimize $w_i$ based on continuous approximation of logit AOPC (referred to as $DBanzhaf(MoRF)$).

**Models and datasets.** We train XGBoost classifiers for several real world tabular datasets (Bank Marketing (Moro & Cortez, 2014), Communities and Crime (Redmond, 2002), Adult Census (Becker & Kohavi, 1996), Diabetes (Kahn), and California Housing (Pace & Barry, 1997)). We use the default settings provided by the training package. Each model is trained on an random 80% split of the corresponding dataset. We also train modified ResNet101 classifiers on Imagenette and Imagewoof datasets (Howard & Gugger, 2020). Details on the models and hardware, as well as simple input-label descriptions of the datasets are provided in Appendix B.

**Evaluation.** We evaluate the faithfulness of the attributions using AOPC with predicted class's logit (Logit AOPC) and probability (Probability AOPC), as well as Iterative Removal of Features (IROF) (Rieger & Hansen, 2020). Higher AOPC and lower IROF represents greater fidelity. It should be noted that while there are discussions on biases with these metrics (Hooker et al., 2019; Rong et al., 2022; Wang & Wang, 2024), they are still widely used in explainable AI literature for evaluation and it is outside of the scope of the study to discuss their limitations.

We evaluate the sensitivity of the attributions using $L_2$-normalized error following Liu et al. (2024), average pairwise rank correlation, and top-$K$ Jaccard index. For The last metric, we set $K$ to 5 for Adult and Diabetes datasets, and the minimum between 20 and half of the number of features for the rest of the datasets. Detailed descriptions of all metrics are provided in Appendix A.

**Settings.** For the California Housing and Diabetes datasets, which have only $d = 7$ features, we generate explanations with sample size equal to 25%, 50%, and 100% of $2^d = 128$. For the rest of the tabular datasets, we use 500 to 1000 perturbations at 100 intervals to evaluate the explanations. The attribution is performed across 40 different seeds between 0 and 800. The replacement value for masking is a random instance in the opposite class. For image datasets, we use a baseline of 0 with a fixed seed of 0. The images are segmented into 64 square segments. All evaluations are performed on the remaining 20% test split.

### 4.2 FAITHFULNESS AND SENSITIVITY ANALYSIS

#### 4.2.1 FAITHFULNESS

The faithfulness evaluations on Bank Marketing dataset are shown in Figure 3. The complete results are reported Appendix D and E in both table and graph format. Across the datasets, we observe several patterns:

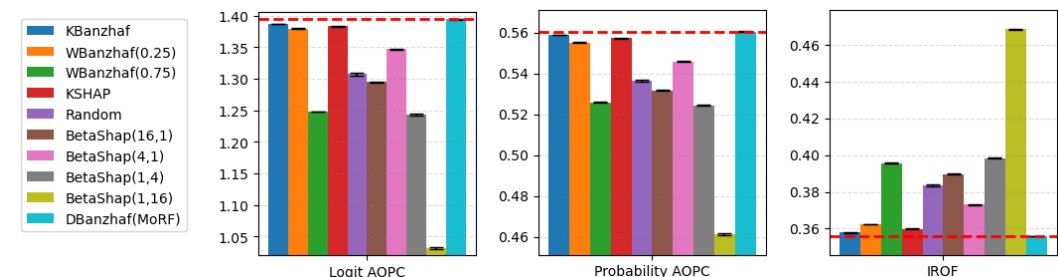

Figure 3: Faithfulness results for Bank Marketing dataset. The red line marks the level of $DBanzhaf(MoRF)$ (rightmost bar). For all three measures, $DBanzhaf(MoRF)$ makes statistically significant improvements in fidelity.

- The average faithfulness is generally the highest for $DBanzhaf(MoRF)$. For several datasets, the difference is statistically significant compared to the standard error.

- The only times where $DBanzhaf(MoRF)$ shows non-significant difference in fidelity is when it shows comparable performance to $KBanzhaf$. For such scenarios, it is likely $w_i = 0.5 \quad \forall i$ is optimal or near-optimal probabilities already.

- For other methods, the ranks can vary across dataset and metric. However, Banzhaf variants (effectively special cases of $DBanzhaf$) tend to outperform non-Banzhaf methods most of the time.

- Using random $w_i$ ($Random$) results in degraded performance. This decrease in fidelity suggests that randomly choosing masking probabilities is worse than using a fixed probability across all features for fidelity in terms of the chosen metrics.

Overall, these results indicate that we can achieve greater average faithfulness by optimizing $\mathbf{w}$ on continuous approximation of AOPC, highlighting the importance of careful feature-wise adjustment of masking probabilities to align with user criteria.

### 4.2.2 SENSITIVITY

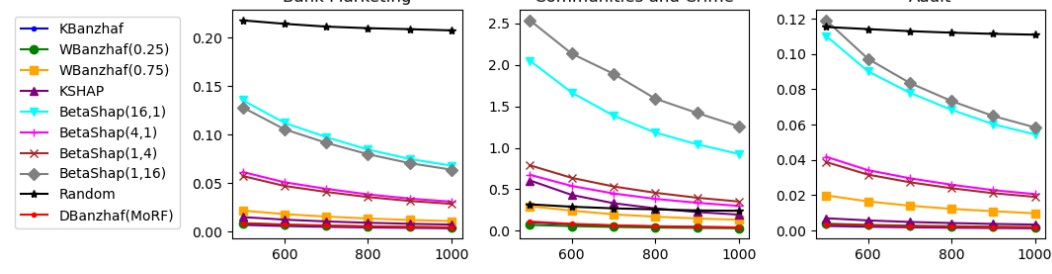

Figure 4: $L_2$-normalized error over $n$ across different datasets. We see that $KBanzhaf$ generally achieves the lowest sensitivity, followed closely by $WBanzhaf(\alpha)$ and $DBanzhaf(MoRF)$.

The $L_2$-normalized error for datasets with large number of features (Bank Marketing, Communities and Crime, and Adult) is presented in Figure 4. Other results (including other metrics and the results for Diabetes and California Housing datasets) are reported in Appendix D and F. We observe the following patterns:

- $KBanzhaf$ tends to achieve the lowest sensitivity in terms of $L_2$-normalized error among all Banzhaf variants across almost all datasets, which supports Theorem 2.

Table 1: Average faithfulness for Dynamic Banzhaf value with masking probability focus placed on image center versus image periphery.

| Name | Logit AOPC | Probability AOPC | IROF |
|------|-----------|------------------|------|
| $DBanzhaf(Center)$ | 5.0321 | 0.7224 | 0.2049 |
| $DBanzhaf(Periph)$ | 4.8623 | 0.7154 | 0.2113 |

(a) Imagenette

| Name | Logit AOPC | Probability AOPC | IROF |
|------|-----------|------------------|------|
| $DBanzhaf(Center)$ | 5.2139 | 0.7388 | 0.1413 |
| $DBanzhaf(Periph)$ | 5.0047 | 0.7310 | 0.1482 |

(b) Imagewoof

- $DBanzhaf(MoRF)$ shows low sensitivity as well, performing nearly as good as $KBanzhaf$ in most scenarios. In contrast, $Random$ performs poorly in all scenarios, showing a clear difference between random and proper $w_i$ selection.

- $KSHAP$ achieves lower sensitivity than $WBanzhaf(0.75)$ in most datasets, while $BetaSHAP$ all have significantly higher $L_2$-normalized errors.

- The results using other sensitivity metrics generally agree with the trends in $L_2$-normalized error with some variations. Most notably, for Jaccard distance and correlation, $BetaShap$ variants can sometimes outperform the Banzhaf variants.

Overall, the results indicate that while Banzhaf variants approach their actual values in a more stable manner (lower $L_2$-normalized error), the ranking stability can vary (mixed Jaccard distance and average pairwise rank correlation). Combined with the fidelity evaluation, we can surmise that:

- $KBanzhaf$ (setting $w_i = 0.5 \quad \forall i$) is generally the best for sensitivity in terms of $L_2$-normalized error.

- On average, $DBanzhaf(MoRF)$ can provide more accurate (or at least equivalently accurate) attribution compared to other Banzhaf variants at the cost of slight deterioration in sensitivity.

- Using random $w_i$ performs worst in terms of sensitivity. Combined with the prior two points, this result again highlights the importance of methodical mask probability adjustment: arbitrarily chosen **w** can be detrimental to attribution quality, while carefully calibrated probabilities can improve those properties.

- $BetaShap$ variants are 'consistently less right' - they sometimes have lower sensitivity than Banzhaf variants, but still have lower average fidelity.

### 4.3 Understanding The Role of Masking Probabilities in Fidelity

In this section, we apply two different $w_i$ between the center and periphery of images in the image datasets to investigate the information captured by $w_i$. Specifically, we compare the faithfulness of explanations depending on the location of high and low $w_i$ in the image. For each image, we divide the image into 64 equally sized segments. Using $w_{high} = 0.7$ (high probability) and $w_{low} = 0.3$ (low probability), we set the masking probability of either the center $4 \times 4$ segments ($DBanzhaf(Center)$) or the periphery segments ($DBanzhaf(Periph)$) to $w_{low}$. The opposite set of segments are set to $w_{high}$. Comparing the faithfulness between the two setups, we find that $DBanzhaf(Center)$ has higher average fidelity than $DBanzhaf(Periph)$ (Table 1). Given that many instances in the image datasets have their objects at the center of the image, this result implies that a higher overlap between the object and low $w_i$ tends to result in more faithful attributions on

average. In other words, $w_i$ seems to act as 'prior information regarding feature importance': by assigning lower $w_i$ to clearly important features (the objects of an image), they are removed more frequently so that more accurate ranking can be computed for the remaining inputs.

## 5 CONCLUSION

In this paper, we present Dynamic Banzhaf, an axiomatic feature attribution method that computes feature importance with each feature having a unique probability of joining a coalition (i.e., the feature masking probability). We prove that Dynamic Banzhaf values can be computed through a centered linear regression and derive the theoretical convergence given a set of masking probabilities. We also discuss calculating optimal probabilities through continuous objective functions that represent user criteria. We optimize the probabilities of Dynamic Banzhaf on a continuous approximation of a faithfulness metric, and compare its performance in terms of fidelity and sensitivity against other game-theoretic attribution methods. We find that Dynamic Banzhaf value meets or beats most baseline methods in terms of average fidelity across all datasets with minimal degradation in sensitivity. These results indicate the importance of adjusting masking probabilities with appropriate objective function to improve the quality of explanations in machine learning.

Given that game-theoretic explainable AI is becoming important for other tasks such as data valuation, it may be interesting to see if we can generalize Dynamic Banzhaf to different applications. In particular, discovering appropriate objective functions for different tasks and metrics could help users gain greater understanding into a model's behavior. Another interesting future research direction would be extending this work to other data, especially for those that need to reflect the original data probability in the masking process.

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

## A    FAITHFULNESS AND SENSITIVITY METRICS

In explainable AI, the explanations should satisfy several desiderata (Nauta et al., 2023). Two properties often discussed in the literature are faithfulness and sensitivity. Faithfulness (also known as fidelity) is the degree to which an explanation matches the model output behavior. Ideally, more important features would have greater impact on model decisions (Hedström et al., 2023). Sensitivity (also known as robustness or consistency) measures the stability of the attribution for identical inputs (Nauta et al., 2023). A good explanation should be stable, i.e., have lower sensitivity. We can quantify these methods using several metrics, which we discuss below.

### A.1    FAITHFULNESS

We consider two traditional metrics of faithfulness: Area over Perturbation Curve (AOPC) (Tomsett et al., 2020) and Iterative Removal of Features (IROF) (Rieger & Hansen, 2020). AOPC measures faithfulness by measuring the area over the curve (AOC) of the change in model output from the original as we replace each feature in the input with the chosen baseline:

$$AOPC = \frac{1}{T+1} \sum_{k=1}^{T} (f(x_R^0) - f(x_R^k)) \tag{7}$$

Where $R$ is the feature replacement procedure and $T$ is the number of replacements. For classifiers, $f$ is usually the logit or probability of the target class. IROF follows a similar process, except that it works exclusively with probabilities:

$$IROF = \frac{1}{T+1} \sum_{k=1}^{T} (p(x_R^k)/p(x_R^0)) \tag{8}$$

A model is more faithful if it has higher AOPC and lower IROF.

### A.2    SENSITIVITY

For sensitivity, we evaluate $L_2$-normalized error (Liu et al., 2024) between attributions across several runs and the true value. If the true value is difficult to calculate, we use the average as the proxy.

$$err_{l2} = \frac{1}{|\bar{\boldsymbol{\beta}}|_2^2} \sum_{k=1}^{T} |\boldsymbol{\beta}_k - \bar{\boldsymbol{\beta}}|_2^2 \tag{9}$$

We also use average pairwise rank correlation and top-$K$ Jaccard distance among multiple evaluations (Nauta et al., 2023).

$$\bar{\rho} = \frac{1}{T-1} \sum_{k=1}^{T-1} \rho(\boldsymbol{\beta}_k, \boldsymbol{\beta}_{k+1}) \tag{10}$$

$$\bar{J}_K = \frac{1}{T-1} \sum_{k=1}^{T-1} \left(1 - \frac{|topk(\boldsymbol{\beta}_k) \cap topk(\boldsymbol{\beta}_{k+1})|}{|topk(\boldsymbol{\beta}_k) \cup topk(\boldsymbol{\beta}_{k+1})|}\right) \tag{11}$$

where $L$ is the number of times the explanation has been evaluated. A method is less sensitive if it has lower $L_2$-normalized error, higher correlation, and higher Jaccard index.

### A.3    A CONTINUOUS APPROXIMATION OF AOPC

While it is possible to optimize masking probabilities $\{\mathbf{w}\}_{i=1,...,n}$ directly on AOPC, it can be computationally intensive since AOPC is discrete and requires recomputing $f(x_R^k)$ at each optimization

step. However, we may approximate streamline the optimization by approximating AOPC with the weighted sum of Dynamic Banzhaf values. Assume the special case where the game $v$ is additive. Then, letting $T = d$:

$$AOPC_{\psi} = \frac{1}{d+1} \sum_{k=1}^{d} (v(\mathbf{z}_R^0) - v(\mathbf{z}_R^k)) = \frac{1}{d+1} \sum_{k=1}^{d} \sum_{j=1}^{k} \psi_{\pi(j)} = \frac{1}{d+1} \sum_{k=1}^{d} (d - k + 1) \psi_{\pi(k)} \tag{12}$$

where $\pi(j)$ is $j$-th feature based on ranking $\pi$. We can approximate $f(\mathbf{z}_R^0) - f(\mathbf{z}_R^k)$ with $\sum_{j=1}^{k} \psi_{\pi(j)}$ because $\mathbf{z}$ is binary, so removal is equivalent to setting $z_i = 0$. Since $AOPC_{\psi}$ is a continuous function with respect $\psi$, which in turn is continuous with respect to $\mathbf{w}$, we can optimize the sum directly using gradient-based methods. In the general case where $v$ is not additive, the difference between $AOPC$ and $AOPC_{\psi}$ is bounded by $\epsilon_{max}$, the maximum difference between $v(\mathbf{z})$ and $\sum_{j, z_j=1} \psi_j$:

$$|AOPC - AOPC_{\psi}| \leq 2\epsilon_{max} \tag{13}$$

Lastly, it should be noted that since we generally cannot compute the Dynamic Banzhaf values $\psi$ analytically, we use the kernelized approximation $\beta_n^*$ instead. This replacement results in the formula in Equation 6.

## B   Experimental Details

The XGBoost classifiers in the experiments are trained with default parameters from the xgboost package, while the image classifiers are fine-tuned from IMAGENET10K weight available in the torchvision package. The classification block of the image classifiers consist of 4 linear layers with 20% dropout, batch normalization, and ReLU activation. All training and experiments are performed on Intel(R) Xeon(R) Gold 6342 CPU @ 2.8GHz and NVidia RTX A6000 (48GB). Input-label descriptions of the datasets are provided in Table 4.

Table 2: Model details.

| Dataset | Model | Package | Acc (%) |
|---|---|---|---|
| Adult | XGBoost | xgboost | 87.29 |
| California Housing | XGBoost | xgboost | 84.74 |
| Communities and Crime | XGBoost | xgboost | 80.75 |
| Imagenette | ResNet101 | torchvision | 89.81 |
| Imagewoof | ResNet101 | torchvision | 79.89 |

Table 3: MLP classification block details.

| Block | Layer Type |
|---|---|
| 1 | ReLU |
| 1 | Linear(2048,1024) |
| 1 | BatchNorm |
| 2 | ReLU |
| 2 | Dropout(0.2) |
| 2 | Linear(1024,512) |
| 2 | BatchNorm |
| 3 | ReLU |
| 3 | Dropout(0.2) |
| 3 | Linear(512,256) |
| 3 | BatchNorm |
| 4 | ReLU |
| 4 | Dropout(0.2) |
| 4 | Linear(512,10) |

Table 4: Dataset description.

| Dataset | Description |
|---|---|
| Bank Marketing | Input: client information. Label: client subscribes to a term deposit |
| Communities and Crime | Input: county information. Label: crime rate is above median |
| Adult | Input: census information. Label: annual income of above $50K |
| Diabetes | Input: county information. Label: crime rate is above median |
| California Housing | Input: housing factors. Label: housing price is above median |
| Imagenette | Input: images from 10 classes in Imagenet. Label: class labels |
| Imagewoof | Input: images from 10 dog breeds in Imagenet. Label: class labels |

## C    PROOFS

In this section, we present the full proofs for theorems 1 and 2

### C.1    PROOF FOR THEOREM 1

Expanding Equation 5, we have:

$$E[(v(\mathbf{z}) - \beta_0 - \boldsymbol{\beta}^T \mathbf{z})^2]$$

$$= E[(v(\mathbf{z}) - \beta_0 - \sum_{i=1}^{d} \beta_i z_i)^2]$$

$$= E[v^2 - 2v \sum_{i=1}^{d} \beta_i z_i + \sum_{i=1}^{d} \sum_{j=1}^{d} \beta_i z_i \beta_j z_j + \beta_0^2 - 2\beta_0 v + 2\beta_0 \sum_{i=1}^{d} \beta_i z_i]$$

$$= E[v^2 - 2v \sum_{i=1}^{d} \beta_i z_i + \sum_{i=1}^{d} \beta_i^2 z_i^2 + \sum_{i \neq j}^{d} \beta_i \beta_j z_i z_j + \beta_0^2 - 2\beta_0 v + 2\beta_0 \sum_{i=1}^{d} \beta_i z_i]$$

$$= E[(1-d)v^2 + \sum_{i=1}^{d} (v - \beta_i z_i)^2 + \sum_{i \neq j}^{d} \beta_i \beta_j z_i z_j + \beta_0^2 - 2\beta_0 v + 2\beta_0 \sum_{i=1}^{d} \beta_i z_i]$$

$$= (1-d)E[v^2] + \sum_{i=1}^{d} E[(v - \beta_i z_i)^2] + \sum_{i \neq j}^{d} \beta_i \beta_j E[z_i z_j] + \beta_0^2 - 2\beta_0 E[v] + 2\beta_0 \sum_{i=1}^{d} \beta_i E[z_i] \tag{14}$$

Since centering sets $E[z_i] = 0$ and $E[z_i z_j] = 0$:

$$\boldsymbol{\psi} = arg \min_{\boldsymbol{\beta}}[(1-d)E[v^2] + \sum_{i=1}^{d} E[(v - \beta_i z_i)^2] + \beta_0^2 - 2\beta_0 E[v]] = arg \min_{\boldsymbol{\beta}}[\sum_{i=1}^{d} E[(v - \beta_i z_i)^2]] \tag{15}$$

which is equivalent to minimizing $\psi_i$ individually. Taking the derivative for a single $\psi_i$, we have:

$$\frac{dE[(v - \beta_i z_i)^2]}{d\beta_i} = E[-2z_i(v - \beta_i z_i)] = 0 \tag{16}$$

$$\rightarrow \beta_i = E[z_i v]/E[z_i^2]$$

Since $E[z_i^2] = Var(z_i) = w_i(1 - w_i)$ and $E[z_i v] = w_i(1 - w_i)E[v|z_i = 1 - w_i] + (1 - w_i)(-w_i)E[v|z_i = -w_i]$:

$$\beta_i = \frac{w_i(1 - w_i)E[v|z_i = 1 - w_i] + (1 - w_i)(-w_i)E[v|z_i = -w_i]}{w_i(1 - w_i)} \tag{17}$$

$$= E[v|z_i = 1 - w_i] - E[v|z_i = -w_i]$$

Since $z_i = 1 - w_i$ means feature $i$ is included in the input set $S$ and $z_i = -w_i$ means it is excluded from $S$, the above equation becomes:

$$\beta_i = E[v(i \cup S)] - E[v(S)]$$

$$= \sum_{S \subseteq N \setminus i} [\prod_{j \in S} w_j \prod_{j \notin S} (1 - w_j)][v(S \cup i)] - \sum_{S \subseteq N \setminus i} [\prod_{j \in S} w_j \prod_{j \notin S} (1 - w_j)][v(S)] \tag{18}$$

$$= \sum_{S \subseteq N \setminus i} [\prod_{j \in S} w_j \prod_{j \notin S} (1 - w_j)][v(S \cup i) - v(S)] = \psi_i$$

## C.2 Proof for Theorem 2

This proof closely follows the convergence of GLIME (Tan et al., 2024). Since Dynamic Banzhaf value is the solution for a linear regression model, we know that:

$$\psi = (Z_n^T Z_n)^{-1} Z_n V_n \tag{19}$$

where $Z_n$ is the centered sampled masks and $V_n$ is the corresponding model predictions. Representing $\Sigma_n = Z_n^T Z_n$ and $\Gamma_n = Z_n V_n$, we would like to find the convergence of $\Sigma_n^{-1}\Gamma_n$ to the limit $\Sigma^{-1}\Gamma$.

First, we can find the limit for $\Sigma_n$ as:

$$\Sigma = \lim_{n \to \infty} \Sigma_n = \lim_{n \to \infty} Z_n^T Z_n = E(Z^T Z) = Var(Z) = diag(\sigma_i^2) = diag(w_i(1-w_i)) \tag{20}$$

$E(Z^T Z)$ is equal to the variance of $Z$ since $Z$ has been centered, i.e., $E(z_i) = 0 \quad \forall i$, which makes $Cov(z_i, z_j) = E(z_i z_j) - E(z_i)E(z_j) = E(z_i z_j)$. Note that $0 \leq \sigma_i^2 \leq 0.25$ since each mask follows a Bernouilli distribution. We can also bound the values of $\Sigma_n$ as follows:

$$\hat{\sigma_n^i} = \frac{1}{n}\{\sum_{k \in S_1} w_i^2 + \sum_{k \in S_2}(1-w_i)^2\} \leq \frac{1}{n}\sum_{k=1}^{n} max(w_i, 1-w_i)^2 \tag{21}$$

$$\hat{\sigma_n}^{ij} = \frac{1}{n}\{\sum_{k \in S_1} w_i w_j + \sum_{k \in S_2} -w_i(1-w_j)$$

$$+ \sum_{k \in S_3} -(1-w_i)w_j + \frac{1}{n}\sum_{k \in S_4}(1-w_i)(1-w_j)\} \tag{22}$$

$$\leq \frac{1}{n}\sum_{k=1}^{n} max(w_i w_j, (1-w_i)(1-w_j) \leq 1$$

Therefore, all elements of $||\Sigma_n - \Sigma||$ are bounded to $[-0.25, 1]$, and we may apply matrix Hoeffding's inequality with $\sigma^2 = max(\sigma_i^2)$:

$$P(||\Sigma_n - \Sigma||_2 \geq t) \leq 2d exp\left(-\frac{nt^2}{8\sigma^2}\right) \tag{23}$$

$||\Sigma^{-1}||_F^2$ is simply the sum of inverse of variances $\sum_d 1/\sigma_i^2 = \gamma^2$. Lastly, we may apply Hoeffding's inequality to $\Gamma_n$ to find:

$$P(||\Gamma_n - \Gamma||_2 \geq t) \leq 2d exp\left(-\frac{nt^2}{8M^2 d^2}\right) \tag{24}$$

Following Tan et al. (2024), if we let $n$ be the maximum among $n_1 = 32\gamma^2\sigma^2 log(4d/\delta)$, $n_2 = 32\epsilon^-2 M^2 d^2 \gamma^2 log(4d/\delta)$, and $n_3 = 32\epsilon^-2 M^2 \sigma^2 d\gamma^4 log(4d/\delta)$, we have $P(||\Sigma_n^{-1}\Gamma_n - \Sigma^{-1}\Gamma|| \leq 1 - \delta)$.

## D    TABULATED RESULTS

The following tables present faithfulness and sensitivity results for 1000 perturbations for datasets with large number of features (Bank Marketing, Communities and Crime, Adult) and 128 perturbations for datasets with small number of features (Diabetes, California Housing). While not reported, the results for smaller number of perturbations are qualitatively similar.

Table 5: Average faithfulness and standard errors for Bank Marketing dataset

| Name | Logit AOPC | Probability AOPC | IROF |
|---|---|---|---|
| $KBanzhaf$ | 1.3866 ± 0.0004 | 0.5589 ± 0.0001 | 0.3578 ± 0.0001 |
| $WBanzhaf(0.25)$ | 1.3794 ± 0.0004 | 0.5551 ± 0.0001 | 0.3624 ± 0.0001 |
| $WBanzhaf(0.75)$ | 1.2479 ± 0.0007 | 0.5259 ± 0.0002 | 0.3957 ± 0.0002 |
| $KSHAP$ | 1.3831 ± 0.0005 | 0.5572 ± 0.0001 | 0.3598 ± 0.0002 |
| $Random$ | 1.3074 ± 0.0018 | 0.5366 ± 0.0005 | 0.3834 ± 0.0006 |
| $BetaShap(16, 1)$ | 1.2946 ± 0.0008 | 0.5317 ± 0.0002 | 0.3896 ± 0.0003 |
| $BetaShap(4, 1)$ | 1.3474 ± 0.0008 | 0.5459 ± 0.0002 | 0.3730 ± 0.0002 |
| $BetaShap(1, 4)$ | 1.2433 ± 0.0011 | 0.5244 ± 0.0003 | 0.3984 ± 0.0004 |
| $BetaShap(1, 16)$ | 1.0320 ± 0.0013 | 0.4615 ± 0.0004 | 0.4686 ± 0.0005 |
| $DBanzhaf(MoRF)$ | 1.3943 ± 0.0004 | 0.5605 ± 0.0001 | 0.3559 ± 0.0001 |

Table 6: Average faithfulness and standard errors for Communities and Crime dataset

| Name | Logit AOPC | Probability AOPC | IROF |
|---|---|---|---|
| $KBanzhaf$ | 5.6177 ± 0.0043 | 0.8763 ± 0.0002 | 0.0544 ± 0.0002 |
| $WBanzhaf(0.25)$ | 5.5966 ± 0.0037 | 0.8703 ± 0.0002 | 0.0607 ± 0.0002 |
| $WBanzhaf(0.75)$ | 5.1575 ± 0.0104 | 0.8666 ± 0.0006 | 0.0641 ± 0.0006 |
| $KSHAP$ | 5.3053 ± 0.0092 | 0.8562 ± 0.0008 | 0.0746 ± 0.0008 |
| $Random$ | 5.4366 ± 0.0114 | 0.8699 ± 0.0006 | 0.0610 ± 0.0006 |
| $BetaShap(16, 1)$ | 5.0147 ± 0.0144 | 0.8223 ± 0.0015 | 0.1104 ± 0.0015 |
| $BetaShap(4, 1)$ | 5.3727 ± 0.0103 | 0.8508 ± 0.0008 | 0.0807 ± 0.0009 |
| $BetaShap(1, 4)$ | 5.4124 ± 0.0111 | 0.8669 ± 0.0006 | 0.0639 ± 0.0006 |
| $BetaShap(1, 16)$ | 4.8278 ± 0.0179 | 0.8471 ± 0.0011 | 0.0845 ± 0.0011 |
| $DBanzhaf(MoRF)$ | 5.6284 ± 0.0043 | 0.8765 ± 0.0002 | 0.0542 ± 0.0002 |

Table 7: Average faithfulness and standard errors for Adult dataset

| Name | Logit AOPC | Probability AOPC | IROF |
|---|---|---|---|
| $KBanzhaf$ | 2.7221 ± 0.0003 | 0.6432 ± 0.0000 | 0.2648 ± 0.0001 |
| $WBanzhaf(0.25)$ | 2.7230 ± 0.0004 | 0.6417 ± 0.0001 | 0.2666 ± 0.0001 |
| $WBanzhaf(0.75)$ | 2.6663 ± 0.0007 | 0.6370 ± 0.0001 | 0.2718 ± 0.0001 |
| $KSHAP$ | 2.7234 ± 0.0004 | 0.6424 ± 0.0001 | 0.2656 ± 0.0001 |
| $Random$ | 2.6890 ± 0.0013 | 0.6374 ± 0.0002 | 0.2712 ± 0.0002 |
| $BetaShap(16, 1)$ | 2.6800 ± 0.0007 | 0.6329 ± 0.0002 | 0.2766 ± 0.0002 |
| $BetaShap(4, 1)$ | 2.7095 ± 0.0006 | 0.6385 ± 0.0001 | 0.2702 ± 0.0001 |
| $BetaShap(1, 4)$ | 2.6790 ± 0.0008 | 0.6384 ± 0.0001 | 0.2706 ± 0.0001 |
| $BetaShap(1, 16)$ | 2.5865 ± 0.0013 | 0.6248 ± 0.0002 | 0.2859 ± 0.0002 |
| $DBanzhaf(MoRF)$ | 2.7246 ± 0.0003 | 0.6435 ± 0.0000 | 0.2645 ± 0.0001 |

Table 8: Average faithfulness and standard errors for Diabetes dataset

| Name | Logit AOPC | Probability AOPC | IROF |
|------|-----------|------------------|------|
| $KBanzhaf$ | $3.5125 \pm 0.0045$ | $0.7972 \pm 0.0005$ | $0.1745 \pm 0.0005$ |
| $WBanzhaf(0.25)$ | $3.4762 \pm 0.0054$ | $0.7881 \pm 0.0009$ | $0.1839 \pm 0.0010$ |
| $WBanzhaf(0.75)$ | $3.4157 \pm 0.0104$ | $0.7964 \pm 0.0010$ | $0.1752 \pm 0.0011$ |
| $KSHAP$ | $3.4892 \pm 0.0061$ | $0.7919 \pm 0.0011$ | $0.1798 \pm 0.0011$ |
| $Random$ | $3.4324 \pm 0.0147$ | $0.7876 \pm 0.0020$ | $0.1843 \pm 0.0021$ |
| $BetaShap(16, 1)$ | $3.3496 \pm 0.0137$ | $0.7596 \pm 0.0035$ | $0.2134 \pm 0.0036$ |
| $BetaShap(4, 1)$ | $3.4245 \pm 0.0101$ | $0.7752 \pm 0.0024$ | $0.1972 \pm 0.0025$ |
| $BetaShap(1, 4)$ | $3.4372 \pm 0.0109$ | $0.7964 \pm 0.0010$ | $0.1753 \pm 0.0011$ |
| $BetaShap(1, 16)$ | $3.2732 \pm 0.0150$ | $0.7872 \pm 0.0016$ | $0.1849 \pm 0.0016$ |
| $DBanzhaf(MoRF)$ | $3.5150 \pm 0.0048$ | $0.7970 \pm 0.0005$ | $0.1747 \pm 0.0005$ |

Table 9: Average faithfulness and standard errors for California Housing dataset

| Name | Logit AOPC | Probability AOPC | IROF |
|------|-----------|------------------|------|
| $KBanzhaf$ | $4.4382 \pm 0.0007$ | $0.8010 \pm 0.0001$ | $0.1280 \pm 0.0001$ |
| $WBanzhaf(0.25)$ | $4.4203 \pm 0.0010$ | $0.7983 \pm 0.0001$ | $0.1310 \pm 0.0001$ |
| $WBanzhaf(0.75)$ | $4.3462 \pm 0.0020$ | $0.7990 \pm 0.0001$ | $0.1301 \pm 0.0001$ |
| $KSHAP$ | $4.4308 \pm 0.0010$ | $0.8002 \pm 0.0001$ | $0.1288 \pm 0.0001$ |
| $Random$ | $4.3765 \pm 0.0023$ | $0.7958 \pm 0.0002$ | $0.1334 \pm 0.0003$ |
| $BetaShap(16, 1)$ | $4.3148 \pm 0.0026$ | $0.7825 \pm 0.0005$ | $0.1476 \pm 0.0005$ |
| $BetaShap(4, 1)$ | $4.3890 \pm 0.0016$ | $0.7922 \pm 0.0003$ | $0.1373 \pm 0.0003$ |
| $BetaShap(1, 4)$ | $4.3873 \pm 0.0015$ | $0.7997 \pm 0.0001$ | $0.1294 \pm 0.0001$ |
| $BetaShap(1, 16)$ | $4.2640 \pm 0.0026$ | $0.7950 \pm 0.0002$ | $0.1346 \pm 0.0002$ |
| $DBanzhaf(MoRF)$ | $4.4375 \pm 0.0008$ | $0.8010 \pm 0.0001$ | $0.1280 \pm 0.0001$ |

Table 10: Average sensitivity for Bank Marketing dataset

| Name | Jaccard | $\bar{\rho}$ | $L_2$ Error |
|------|---------|--------------|-------------|
| $KBanzhaf$ | $0.2236$ | $0.8797$ | $0.0033$ |
| $WBanzhaf(0.25)$ | $0.2107$ | $0.8782$ | $0.0042$ |
| $WBanzhaf(0.75)$ | $0.2880$ | $0.8461$ | $0.0106$ |
| $KSHAP$ | $0.2318$ | $0.8658$ | $0.0071$ |
| $BetaShap(16, 1)$ | $0.0506$ | $0.9645$ | $0.0679$ |
| $BetaShap(4, 1)$ | $0.0592$ | $0.9640$ | $0.0307$ |
| $BetaShap(1, 4)$ | $0.0986$ | $0.9408$ | $0.0285$ |
| $BetaShap(1, 16)$ | $0.1005$ | $0.9355$ | $0.0637$ |
| $Random$ | $0.3463$ | $0.7374$ | $0.2073$ |
| $DBanzhaf(MoRF)$ | $0.2261$ | $0.8765$ | $0.0040$ |

Table 11: Average sensitivity for Communities and Crime dataset

| Name | Jaccard | $\bar{\rho}$ | $L_2$ Error |
|---|---|---|---|
| $KBanzhaf$ | 0.2424 | 0.8106 | 0.0389 |
| $WBanzhaf(0.25)$ | 0.2315 | 0.8105 | 0.0314 |
| $WBanzhaf(0.75)$ | 0.4154 | 0.6459 | 0.1328 |
| $KSHAP$ | 0.4545 | 0.6052 | 0.1951 |
| $BetaShap(16,1)$ | 0.5472 | 0.7634 | 0.9257 |
| $BetaShap(4,1)$ | 0.4120 | 0.8252 | 0.3002 |
| $BetaShap(1,4)$ | 0.4314 | 0.7914 | 0.3527 |
| $BetaShap(1,16)$ | 0.5284 | 0.7458 | 1.2574 |
| $Random$ | 0.4159 | 0.6722 | 0.2414 |
| $DBanzhaf(MoRF)$ | 0.2523 | 0.8004 | 0.0415 |

Table 12: Average sensitivity for Adult dataset

| Name | Jaccard | $\bar{\rho}$ | $L_2$ Error |
|---|---|---|---|
| $KBanzhaf$ | 0.2030 | 0.8753 | 0.0013 |
| $WBanzhaf(0.25)$ | 0.2129 | 0.8678 | 0.0019 |
| $WBanzhaf(0.75)$ | 0.2537 | 0.8412 | 0.0097 |
| $KSHAP$ | 0.2236 | 0.8618 | 0.0033 |
| $BetaShap(16,1)$ | 0.0405 | 0.9682 | 0.0544 |
| $BetaShap(4,1)$ | 0.0408 | 0.9724 | 0.0206 |
| $BetaShap(1,4)$ | 0.0515 | 0.9685 | 0.0190 |
| $BetaShap(1,16)$ | 0.0613 | 0.9591 | 0.0584 |
| $Random$ | 0.2784 | 0.7985 | 0.1110 |
| $DBanzhaf(MoRF)$ | 0.2079 | 0.8717 | 0.0016 |

Table 13: Average sensitivity for Diabetes dataset

| Name | Jaccard | $\bar{\rho}$ | $L_2$ Error |
|---|---|---|---|
| $KBanzhaf$ | 0.0815 | 0.9657 | 0.0043 |
| $WBanzhaf(0.25)$ | 0.0986 | 0.9569 | 0.0063 |
| $WBanzhaf(0.75)$ | 0.1672 | 0.9055 | 0.0272 |
| $KSHAP$ | 0.1214 | 0.9373 | 0.0093 |
| $BetaShap(16,1)$ | 0.0953 | 0.9009 | 0.1193 |
| $BetaShap(4,1)$ | 0.0901 | 0.9279 | 0.0539 |
| $BetaShap(1,4)$ | 0.1084 | 0.9262 | 0.0455 |
| $BetaShap(1,16)$ | 0.1297 | 0.9015 | 0.1113 |
| $Random$ | 0.2028 | 0.8485 | 0.1373 |
| $DBanzhaf(MoRF)$ | 0.0901 | 0.9601 | 0.0052 |

Table 14: Average sensitivity for California Housing dataset

| Name | Jaccard | $\bar{\rho}$ | $L_2$ **Error** |
|---|---|---|---|
| $KBanzhaf$ | 0.0923 | 0.9673 | 0.0024 |
| $WBanzhaf(0.25)$ | 0.1248 | 0.9481 | 0.0058 |
| $WBanzhaf(0.75)$ | 0.2088 | 0.8833 | 0.0178 |
| $KSHAP$ | 0.1285 | 0.9442 | 0.0044 |
| $BetaShap(16,1)$ | 0.0859 | 0.9349 | 0.1126 |
| $BetaShap(4,1)$ | 0.0805 | 0.9526 | 0.0466 |
| $BetaShap(1,4)$ | 0.1096 | 0.9435 | 0.0459 |
| $BetaShap(1,16)$ | 0.1346 | 0.9165 | 0.1191 |
| $Random$ | 0.2162 | 0.8698 | 0.1104 |
| $DBanzhaf(MoRF)$ | 0.1186 | 0.9538 | 0.0033 |

# E    FAITHFULNESS

Additional faithfulness results are visualized in this section.

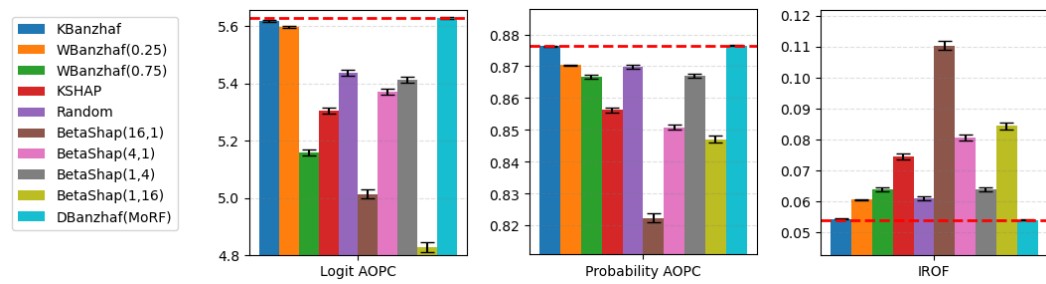

Figure 5: Faithfulness metric visualization for Communities and Crime

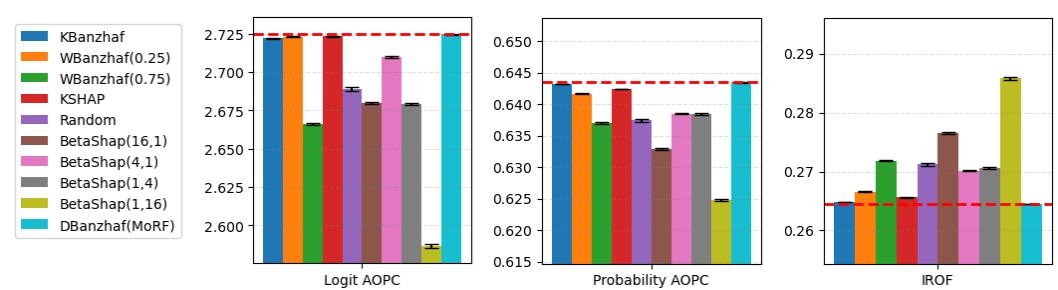

Figure 6: Faithfulness metric visualization for Adult

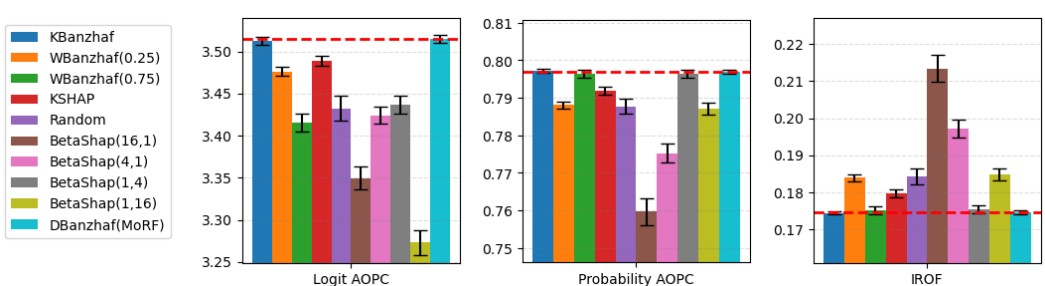

Figure 7: Faithfulness metric visualization for Diabetes.

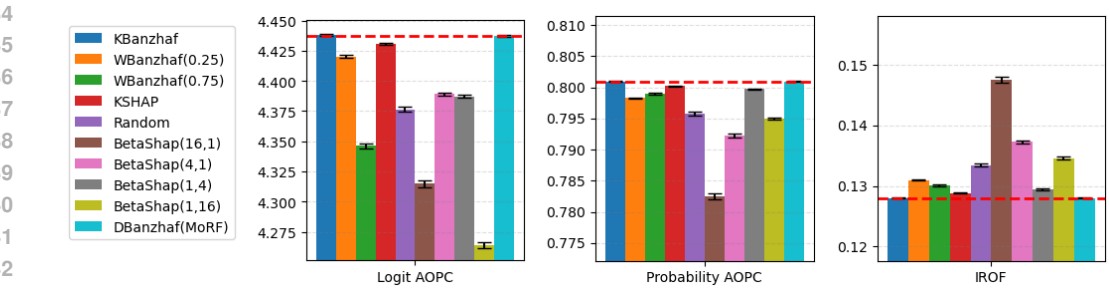

Figure 8: Faithfulness metric visualization to California Housing

# F   SENSITIVITY

The sensitivity results for all datasets are visualized in this section.

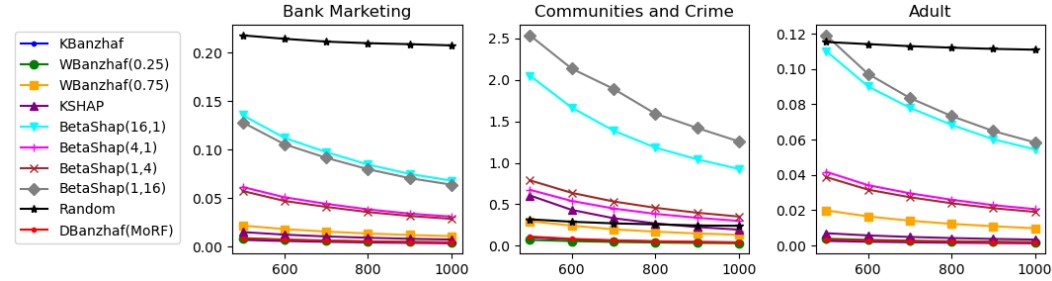

Figure 9: $L_2$-normalized error across datasets with large number of features.

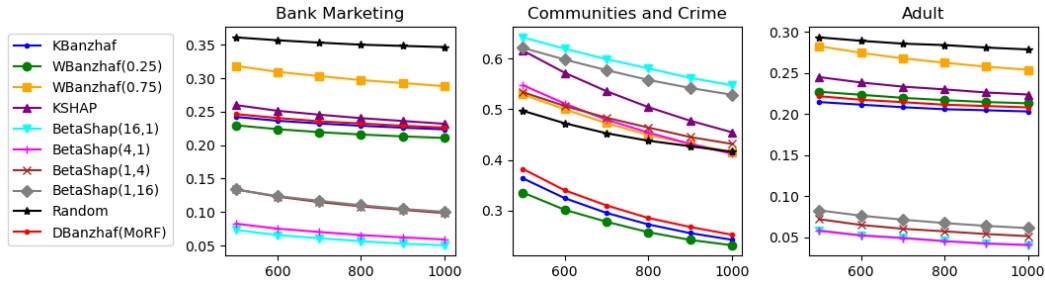

Figure 10: Jaccard distance across datasets with large number of features.

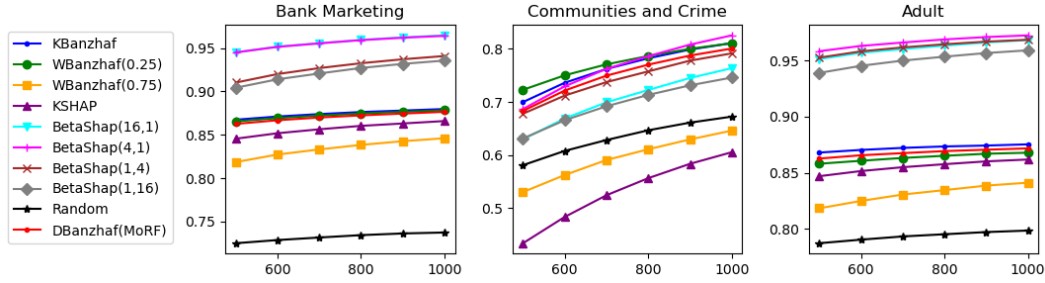

Figure 11: Pairwise correlation across datasets with large number of features.

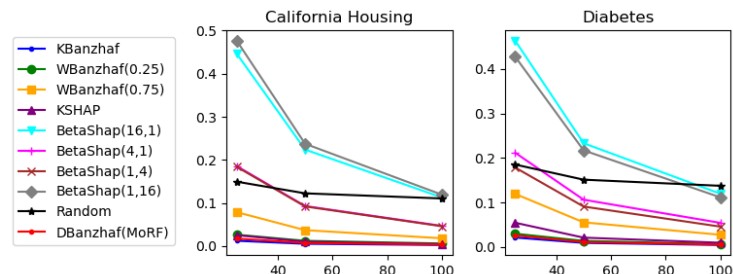

Figure 12: $L_2$-normalized error across datasets with small number of features.

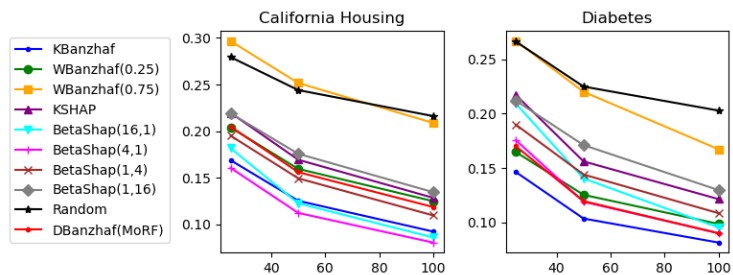

Figure 13: Jaccard distance across datasets with small number of features.

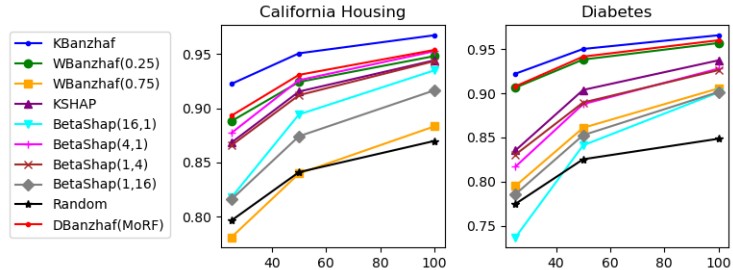

Figure 14: Pairwise correlation across datasets with small number of features.

