# OpenReview forum: "Dynamic Banzhaf: Game-Theoretic Attribution with Dynamic Feature-Wise Probabilities"
_ICLR.cc/2026/Conference — ICLR 2026 Conference Withdrawn Submission_

### Official Review · Reviewer_g4tz · 2025-10-23

**Soundness:** 3
**Presentation:** 2
**Contribution:** 1
**Rating:** 2
**Confidence:** 4

**Summary:**

This paper proposes weighted Banzhaf values (Marichal & Mathonet, 2011) for feature attribution, as an alternative to Banzhaf or Shapley values. The authors propose a data-driven way to find the optimal weights $w_i$ for each feature $i \in N$ by trying to optimize the AOPC metric over all features. The weights are chosen individually for each feature based on this weight, and determine the probability of $i$ occuring in a subset $S \subseteq N$. The authors then use the least-squares representation of weighted Banzhaf values to approximate the attribution values, similar to KernelSHAP or KernelBanzhaf. The method is then benchmarked in terms of faithfulness and sensitivity on different datasets, where marginal improvements are shown over KernelBanzhaf.

**Strengths:**

- The paper is well-written and easy to follow
- User-specific attribution values with clear interpretations are promising
- A data-driven approach to satisfy user-specific needs is a promising direction

**Weaknesses:**

- The theoretical contribution of this paper is not strong: Essentially applying weighted Banzhaf values for feature attribution, and a simple data-driven approach to select "optimal" weights. Theorem 1 directly follows from Marichal & Mathonet (2011) by noticing that the Banzhaf interaction are not affected by constant shifts. For Theorem 2, I presume that a much simpler bound could be found by using leverage scores, see e.g. [1].
- The method only shows marginal improvements over competitors, event though the attribution values were specifically optimized for some metrics: In both failthfulness and sensitivy, KernelBanzhaf almost yields similar results
- centering $z-w$ and fitting an intercept $\beta_0$ most likely decreases performance compared to simply re-defining $\nu_0(S) := \nu(S) - \nu(\emptyset)$. As correctly mentioned by the authors, the weighted Banzhaf interaction values are not affected by those shifts, and thus the exact value for the linear approximation at $\emptyset$ can be included, without the need to fit it separately via intercept with possible error.
- The weighted Banzhaf values directly imply the **best** linear approximation given the specific weights (i.e. the faithfulness metric comparing all subset approximations). Instead of a data-driven approach to maximize the AOPC metric, it would be compelling to find the weights that theoretically optimize this property.


**Minor**
- log-scale in Figure 4 could improve interpretability of the figure
- line 204 $z$ should be $\nu(z)$ (?)

[1] Musco, Christopher, and R. Teal Witter. "Provably Accurate Shapley Value Estimation via Leverage Score Sampling." The Thirteenth International Conference on Learning Representations.

**Questions:**

- Is there any evidence for an **optimal** theoretical choice of weights to maximize the proposed metric?
- Why is it reasonable to choose subsets based on independent probabilities for each feature?

---

### Official Review · Reviewer_LGvL · 2025-10-27

**Soundness:** 2
**Presentation:** 2
**Contribution:** 1
**Rating:** 2
**Confidence:** 4

**Summary:**

The paper presents Dynamic Banzhaf (DBanzhaf) as a new attribution method for aligning the attribution method to pre-defined goals by the user. DBanzhaf is very related to the weighted Banzhaf value but with a dynamic weight (not fixed for all coalitions) attached to each coalition. The dynamic nature of the weights can then be used to optimize the attribution scores to predefined goals.

**Strengths:**

- **Significance**: The work studies an important problem! Particularly for laypersons, the choice of attribution method is not clear. Aligning attribution methods to goals/desiderata in terms of explanation quality (faithfulness, computation efficiency) is a good research direction.
- **Good Connection**: The Banzhaf value is a very game theoretic tool which is still understudied. The link of DBanzhaf (while still under explored) to the weighted Banzhaf value is interesting.

**Weaknesses:**

- **Contribution is under-developed:** The work studies the weighted Banzhaf value (not cited properly) in another way. The DBanzhaf seems to be a generalization of the weighted Banzhaf value where the weights can be "basically" everything. This is interesting from an optimization perspective (see strengths), but is very much open to debate if this is something one would actually want to do. The work does not state what theoretical properties from the Banzhaf value or weighted Banzhaf value are retained in the new DBanzhaf value (I assume none). This must be analyzed and discussed. Otherwise it's quite unclear what the final attribution values actually mean.
- **No Code:** I wanted to double check the implementation of the methods and baselines and noticed that the source code is neither linked nor attached to the submission. The publication of source code is **a necessity to conduct proper peer review**.
- **Empirical Evaluation is inconclusive:** The experiments do not paint a clear picture that doing DBanzhaf compared to the other alternatives is actually a good idea. While the discussion of the faithfulness results presents DBanzhaf as the best method, the plots in Figure 3 (and in the appendix for different datasets) do not really support this claim. DBanzhaf is basically on par with the rest. The same holds true for the evaluation regarding the sensitivity. While the plot in the main paper shows that DBanzhaf has a low sensitivity, the plots in the appendix using different distance metrics shows very different results (DBanzhaf is in the middle of the pack). This basically shows that _sensitivity_ is a rather noisy evaluation metric which does not contain a good signal for proper evaluation.
- **The Related Work of the Paper is not good.** Many important works are missing or wrongfully attributed. For example, the weighted Banzhaf value is attributed to a data valuation paper (Li & Yu, 2024) and the original reference (Marichal & Mathonet, 2011) is only brushed aside as an additional example when talking about technicalities. This is quite sad, since the weighted Banzhaf value is, in my understanding, a special case of the here presented DBanzhaf value.
- **The presentation of the paper is not very good.** The tables and plots are not very clear. For example, the ablation plot in Figure 2 does not have any axis titles and it's hard to know what this plot is supposed to be showing. The Figures in the empirical section (Figure 3 and 4) are also not standard evaluations known in other explainable AI works and are also not very understandable. The work does also not flow well and reads rather like a technical report than a scientific paper.
- **Minor Typos and Grammar:** The writing of the paper can be improved. The manuscript contains a couple of grammatical errors. For example, the first sentence in the abstract be _Game-theoretic attribution methods approximate **a** target model as a cooperative game and evaluate feature importance via payoff allocation to the input features_.
- **Use of many hand-wavy arguments**: The work contains many instances of rather hand-wavy arguments or not well substantiated sentences. For example in lines 73-74 _"Given a model, a user chooses a goal for the attribution, such as improving its quality."_ it is not at all clear what _improving its quality_ could refer to.  This is all left very subjective. There are more than one of these cases in the work.

**Questions:**

**Q1**: What theoretical properties does DBanzhaf still retain from the Banzhaf value or the Weighted Banzhaf value?
**Q2**: In Lines 336-337 you write _"For all three measures, DBanzhaf(MoRF) makes **statistically significant** improvements in fidelity."_ How is the _statistically significance_ checked?
**Q3**: What is Figure 3 **exactly** showing?

---

### Official Review · Reviewer_TvRT · 2025-10-31

**Soundness:** 3
**Presentation:** 3
**Contribution:** 2
**Rating:** 2
**Confidence:** 4

**Summary:**

This manuscript begins with the observation that Shapley's value "applies much higher weights on either very small or very large subsets, ignoring the effects of mid-sized subsets".

From here, it picks up on Weber's work in game theory from the 1970s and 1980s, which noted that Shapley's symmetry assumption could be relaxed to any probability measure.  The authors start with Banzaf's uniform measure, but then tune it to maximise a fidelity measure based on the Area over the Perturbation Curve (AoPC).  They call the result ($\S 3$) a Dynamic Banzhaf (DB) measure.

In $\S 4$, an XGBoost classifier is trained on a number of tabular datasets, and a variety of Shapley and Banzhaf variants, including DB, applied to the result.  Overall, Banzhaf methods seem to outperform non-Banzhaf methods in terms of 'faithfulness' (e.g. replicating the model decisions).  In terms of 'sensitivity' (of model predictions to input perturbations) it is less obvious what the takeaway is.

An experiment is also performing in which either the periphery or the center of an image is given more/less weight by the DB measure.  This finds, as one would expect, that center-weighting yields higher fidelity.

**Strengths:**

Generally, I think that explaining and interpreting large ML models is important - and that the area is in its infancy.  Thus, research in this field is welcome.

Further, the manuscript is well written and soundly executed.

**Weaknesses:**

The manuscript is motivated by a claim that Shapley's weights are somehow problematic - with an implicit assumption that Banzhaf's uniformity is somehow better in general.  While an axiomatization for the DB measure is presented, ultimately the question is not whether an axiomatization exists for a measure (in general, "yes", one will), but whether it corresponds somehow to explainability or interpretability.

From this point of view, the results in $\S 4$, on faithfulness and sensitivity, have to make that case.  I'm not sure, though, that they do:
1. the faithfulness measures seem - if I understand them - to argue that the DB approximation can provide a well-fitting proxy model, not that it is interpretable/explainable.  Further, DB doing well on faithfulness measures should not surprised, as its free parameters were tuned to ensure this.  The interesting question, I think, would be whether there are (general?) lessons that can be extracted from the tuned weights.
1. the sensitivity results seem more mixed re: DB's performance, so it is hard to see how they help the argument that DB aids interpretability/explainability.

Thus, the manuscript feels like a solution looking for a problem: yes, we *can* derive another Shapley variant, but it seems to me that the harder work is working out which of those help us answer model explainability and interpretability questions.  Faithfulness and sensitivity still seem one step removed from that.

Specifically, Shapley's value has a clean interpretation: how much does knowing a feature's specific value change the model's prediction?  I do not see a similarly clean interpretation here.  Thus, I am skeptical that knowing these DB values will help users or developers understand their models better.

Thus, the approach seems to run the risk of 'X hacking' (q.v. Sharma et al. ICML 2025): using the free parameters in an explainability measure to overfit.  Previous work (q.v. Frye, Rowat, Feige ICML 2020) seem more disciplined in their relaxation of symmetry, using it to encode causal knowledge.

On a semantic - rather than substantial - note, it is not clear to me, from the manuscript, that "dynamic Banzhaf" is a good name for this value:
1. if I understand, it is not *dynamic* in the sense that its probability measure varies over time; instead, the measure is tuned.
1. if the Banzhaf index requires a uniform distribution, tuning away from that distribution brings us back to the general class of quasi-values (Weber, 1988).

An even more minor semantic point: the one variant of DB tested is called 'D Banzhaf (MoRF)', which seems more complicated than necessary given that there's no other 'D Banzhaf'.

**Questions:**

How do the authors address the 'X hacking' concerns raised in Sharma et al. (ICML, 2025)?

Are there (general) lessons that can be drawn from how DB sets weights?

---

### Official Review · Reviewer_mxxN · 2025-10-31

**Soundness:** 1
**Presentation:** 1
**Contribution:** 2
**Rating:** 2
**Confidence:** 4

**Summary:**

This paper proposes a new approximator for Banzhaf values, a game-theoretic concept used to estimate feature attributions of machine learning model predictions. Experiments with an XGBoost model and five tabular datasets demonstrate the validity of the approach.

**Strengths:**

The method is quite simple in a good way. The core idea is novel. Unfortunately, this work appears underdeveloped; with proper space management, it could be condensed into 6-7 pages of content.

**Weaknesses:**

This paper primarily references related work from 2015 to 2022, making it generally outdated in light of current research challenges. The motivation and research questions stated here have been studied for the last 10 years to the point where I am doubtful regarding the potential impact of yet another work on the exact same topic. Diving deeper into cooperative game theory to improve a faithfulness metric by a few % (Figures 3 & 4) is not necessarily of interest to learning representations. Note that this is orthogonal to works that actually apply the theory to significantly improve representation learning; see e.g. [a].

Theorems 1 & 2 are trivial extensions of results obtained in related work (conveniently surveyed in [b]).

Experiments omit some of the state-of-the-art approximation algorithms [b].

Figures and Tables are of low quality (visually), making it challenging to interpret the main results.

[a] Hierarchical Banzhaf interaction for general video-language representation learning. CVPR 2023

[b] shapiq: Shapley interactions for machine learning. NeurIPS 2024

**Questions:**

1. How does this method improve the utility of explanations? What is its purpose?

- Writing "Li & Yu (2024) generalizes the Banzhaf value to weighted Banzhaf value" is misleading. They use the weighted Banzhaf value generalized in [c,d].
- The introduction can be greatly condensed, while still conveying the same message.
- Equations (1)-(3) are well-known and can be put inline, saving a lot of white space.
- A key related work is missing [e].

[c] Ding et al. Transforms of pseudo-Boolean random variables. Discrete Applied Mathematics, 2010

[d] Marichal et al. Weighted Banzhaf power and interaction indexes through weighted approximations of games. European Journal of Operational Research, 2011

[e] Faith-Shap: The faithful Shapley interaction index. JMLR 2023

---

### Note · Authors · 2025-11-21

I have read and agree with the venue's withdrawal policy on behalf of myself and my co-authors.